# On Causal Rationalization

Wenbo Zhang[1], Tong Wu[2], Yunlong Wang[2], Yong Cai[2], and Hengrui Cai[1]

[1]Department of Statistics, University of California Irvine
[2]Advanced Analytics, IQVIA

## Abstract

With recent advances in natural language processing, rationalization becomes an essential self-explaining diagram to disentangle the black box by selecting a subset of input texts to account for the major variation in prediction. Yet, existing association-based approaches on rationalization cannot identify true rationales when two or more rationales are highly intercorrelated, and thus provide a similar contribution to prediction accuracy, so-called spuriousness. To address this limitation, we novelly leverage two causal desiderata, non-spuriousness and efficiency, into rationalization from the causal inference perspective. We formally define the probability of causation in the rationale model with its identification established as the main component of learning necessary and sufficient rationales. The superior performance of our causal rationalization is demonstrated on real-world review and medical datasets with extensive experiments compared to state-of-the-art methods.

## 1 Introduction

Recently, large language models (LLMs) have drawn increasing attention and have been widely used in extensive Natural Language Processing (NLP) tasks [see e.g., 22, 6]. Although those deep learning based models could provide incredibly great performance, it remains a daunting task in finding trustworthy explanations to interpret these models' behavior, which is particularly critical in high-stakes applications such as healthcare and finance. In healthcare, longitudinal electronic health records (EHR) are increasingly used to forecast patients' disease progression and assist in clinicians' decision making. Beyond simply predicting clinical outcomes, doctors are more interested in understanding the decision making process of predictive models thereby building trust, as well as extracting clinically meaningful and relevant insights [24]. For example, when developing computational phenotyping algorithms (the process of identifying patients from EHR with certain phenotypic characteristics), sequential data mining methods are popularly used to extract temporal text sequences from EHR data that are working as abstracts or milestones of patient medical journey [16, 38]. Such temporal sequences can oftentimes better characterize patient medical conditions, as raw EHR data are by nature heterogeneous, sparse, and noisy.

Disentangling the black box in deep neural networks is a notoriously challenging task [1]. There are a lot of research works focused on providing trustworthy explanations for models, generally classified into post-hoc techniques and self-explaining models [12, 29]. To provide better model interpretation, self-explaining models are of greater interest, and selective rationalization is one popular type of such a model by highlighting important tokens among input texts [see e.g., 14, 27, 21, 34, 34, 7]. The general framework of selective rationalization as shown in Figure 1a consists of two components, a selector and a predictor. Those selected tokens by the trained selector are called rationales and they are required to provide similar prediction accuracy to the full input text based on the trained predictor. Besides, they should reflect the model's true reasoning process (faithfulness) [14, 21] and provide a convenient explanation to people (plausibility) [14, 7]. Most of the existing works

2022 Trustworthy and Socially Responsible Machine Learning (TSRML 2022) co-located with NeurIPS 2022.

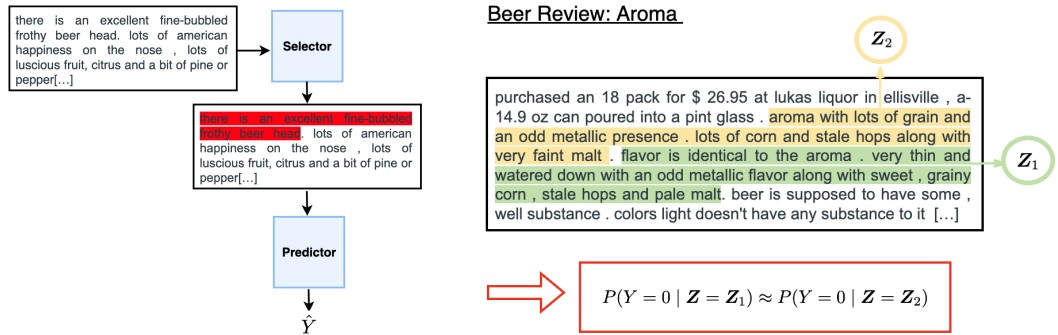

(a) Selective rationalization framework.    (b) Motivating example from the Beer review data.

Figure 1: (a) shows a selective rationalization framework for a beer review, which can be seen as a select-predict pipeline where firstly rationales are selected and then fed into the predictor; (b) provides a motivating example for the Beer review data. Yellow highlights the text related to the smell (aroma) and green highlights the text related to taste. $Y = 0$ indicates a negative sentiment on the smell.

[23, 4, 27, 9] mainly focus on finding rationales ($\boldsymbol{Z}$) which can achieve the best prediction accuracy for the outcome of interest ($Y$) based on input texts as $\boldsymbol{X}$, and thus is association-based models.

The major limitation of these association-based works (also see related works in Appendix A.1) lies in identifying the true rationales when two or more rationales are highly inter-correlated, and thus provide a similar contribution to prediction accuracy. Here, the true rationale, which later we will formally define as the causal rationale, is the true sufficient rationale for the prediction without spurious information that may be related to the outcome of interest but does not indeed cause the outcome. As one example shown in Figure 1b, one chunk of beer review comments covers two aspects, aroma and palate. The reviewer didn't feel it was a good beer in terms of aroma and assigned a low score to it, which can be regarded as negative sentiment. Hence, if we want to explain why the reviewer provided the negative sentiment, the causal interpretation should be the texts highlighted in yellow. However, it can be seen that the texts in terms of the palate, highlighted by green, can provide similar prediction accuracy as the spurious information that is highly correlated with the previous true rationale identified. Therefore, we can't distinguish these two selected snippets when they all have high predictive power. Also sometimes spurious correlations are introduced during training, which misleads the selector to select non-causally related features [9]. Furthermore, if spurious features are selected, the predictor will tend to fail once there is a large discrepancy between the training and testing data distributions and it would be nearly impossible to achieve high generalization performance [31].

In this paper, we propose a novel approach called causal rationalization aiming to find causal interpretations for general NLP tasks. Beyond selecting rationales based on purely prediction performance, our goal is to identify rationales with causal meanings. To achieve this goal, we introduce a novel concept as causal rationales by leveraging two causal desiderata: non-spuriousness and efficiency. This is motivated by a recent work [36] in representation learning, where the two desiderata are adopted in supervised learning tasks. Here non-spuriousness means a representation can capture features causally determining the outcome, and efficiency means only essential and no redundant features are included. Our main contributions are summarized as follows.

• We formally define the probability of causation (POC) for rationales accounting for non-spuriousness and efficiency, and establish identifications of those counterfactual quantities.

• We propose a practically useful algorithm for learning necessary and sufficient rationales that simultaneously estimates POC from observed data and trains the causal rational model, where the existing approach [36] is not applicable.

• The superior performance of our causality-based rationalization method is demonstrated in application on Beer review and Geographic Atrophy (electronic health records) with extensive experiments compared to state-of-the-art methods.

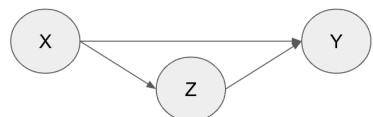

Figure 2: Causal diagram of rationale.

## 2 Framework

Denote $\boldsymbol{X} = (X_1, \cdots, X_d)$ as the text with $d$ tokens, $\boldsymbol{Z} = (Z_1, \cdots, Z_d)$ as the corresponding selection where $Z_i$ is a Bernoulli random variable indicating whether the token $i$ is selected in the rationale and $Y$ as the binary label. Let $Y(\boldsymbol{X} = \boldsymbol{x}, \boldsymbol{Z} = \boldsymbol{z})$ denote the potential value of $Y$ when setting $\boldsymbol{X}$ as $\boldsymbol{x}$, $\boldsymbol{Z}$ as $\boldsymbol{z}$. We generalize three standard assumptions from causal inference literature [28, 35, 20] to rationalization, including consistency, ignorability, and monotonicity. More details can be found in Appendix A.2. A structural causal model (SCM) is defined with a directed acyclic graph (DAG) where nodes are variables and edges represent causal relationships between variables. In this paper, we propose a SCM for rationalization as follows with its DAG shown in Figure 2:

$$\boldsymbol{X} = f(N_X), \quad \boldsymbol{Z} = g(\boldsymbol{X}, N_Z), \quad Y = h(\boldsymbol{Z} \odot \boldsymbol{X}, N_Y), \tag{1}$$

where $N_X, N_Y, N_Z$ are exogenous variables and $f, g, h$ are the causal mechanisms of $\boldsymbol{X}, \boldsymbol{Z}, Y$ respectively, with $\odot$ denoting the element-wise product. Suppose we observe a data point with the text $\boldsymbol{X}$ and binary selections $\boldsymbol{Z}$, rationales can be represented by the event $\{\boldsymbol{X}_i \mathbb{I}(\boldsymbol{Z}_i = 1)\}_{1 \le i \le d}$, where $\mathbb{I}(\boldsymbol{Z}_i = 1)$ indicates if the $i$-th token is selected, $\boldsymbol{X}_i$ is the corresponding text, and $d$ is the length of the text.

**Remark 2.1.** *The data generation process in* (1) *matches many graphical models in previous work [see e.g., 11, 27]. As a motivating example, consider the sentiment labeling process for the Beer review data. The labeler first locates all the important sub-sentences or words which encode sentiment information and marks them. After reading all the reviews, the labeler then goes back to the previously marked text and makes the final judgment on sentiment. In this process, we can regard the first step of marking important locations of words as generating the selection of $\boldsymbol{Z}$ through reading texts $\boldsymbol{X}$. The second step can be seen as combining the selected locations with raw text to generate rationales (equivalent to $\boldsymbol{Z} \odot \boldsymbol{X}$) and then the label $Y$ is generated through a complex decision function $h$.*

Next, we extend the probability of causation [28, 36] to rationales. Define the probability of sufficiency (PS) for rationales as $\mathrm{PS}_{\{\boldsymbol{Z}=\boldsymbol{z}, \boldsymbol{X}=\boldsymbol{x}\}, Y=y} \triangleq P\left(Y(\boldsymbol{Z}=\boldsymbol{z}, \boldsymbol{X}=\boldsymbol{x}) = y \mid \boldsymbol{Z} \neq \boldsymbol{z}, Y \neq y, \boldsymbol{X}=\boldsymbol{x}\right)$, which is the probability of the potential label with rationales $\{\boldsymbol{X}_i \mathbb{I}(\boldsymbol{Z}_i = 1)\}_{1 \le i \le d}$. It can be seen as the capacity of the rationales to "produce" the label. Efficient rationales shall have a high PS. On the other hand, the probability of necessity (PN) for rationales is defined by $\mathrm{PN}_{\{\boldsymbol{Z}=\boldsymbol{z}, \boldsymbol{X}=\boldsymbol{x}\}, Y=y} \triangleq P\left(Y(\boldsymbol{Z} \neq \boldsymbol{z}, \boldsymbol{X}=\boldsymbol{x}) \neq y) \mid \boldsymbol{Z}=\boldsymbol{z}, Y=y, \boldsymbol{X}=\boldsymbol{x}\right)$, which is the probability of the rationales $\{\boldsymbol{X}_i \mathbb{I}(\boldsymbol{Z}_i = 1)\}_{1 \le i \le d}$ being a necessary cause of the label $\mathbb{I}\{Y = y\}$. Non-spurious rationales shall have a high PN. To achieve both non-spuriousness and efficiency as the desired causal rationales, we further define the probability of sufficiency and necessity as follows.

**Definition 2.1.** *Probability of necessity and sufficiency (*PNS*) for rationales:*

$$\mathrm{PNS}_{\{\boldsymbol{Z}=\boldsymbol{z}, \boldsymbol{X}=\boldsymbol{x}\}, Y=y} \triangleq P\left(Y(\boldsymbol{Z} \neq \boldsymbol{z}, \boldsymbol{X}=\boldsymbol{x}) \neq y, Y(\boldsymbol{Z}=\boldsymbol{z}, \boldsymbol{X}=\boldsymbol{x}) = y\right). \tag{2}$$

When both the selection $\boldsymbol{Z}$ and the label $Y$ are univariate binary and $\boldsymbol{X}$ is removed, then our defined PN, PS, and PNS boil-down into the classical definitions of POC in Definition 9.2.1 to Definition 9.2.3 of Pearl (2000) [28]), respectively. In addition, our definitions imply that the input texts are fixed and the probabilities of causation are mainly detected through interventions on selected rationales. This not only reflects the data generation process we proposed in Model (1), but also distinguishes our settings with existing causal frameworks [see e.g., 28, 36]. Since most times we are more interested in efficiency and non-spuriousness of each single rationale, we extend the work of Wang and Jordan (2021) [36] to further generalize PNS to a conditional version with respect to the selections.

**Definition 2.2.** *Conditional probability of necessity and sufficiency (*CPNS*) for $j$th selection:*

$$\mathrm{CPNS}_{\{Z_j=z_j, \boldsymbol{X}=\boldsymbol{x}\}, Y=y | \boldsymbol{Z}_{-j}=\boldsymbol{z}_{-j}} = P\left(Y(Z_j=z_j, \boldsymbol{Z}_{-j}=\boldsymbol{z}_{-j}, \boldsymbol{X}=\boldsymbol{x}) = y, Y(Z_j \neq z_j, \boldsymbol{Z}_{-j}=\boldsymbol{z}_{-j}, \boldsymbol{X}=\boldsymbol{x}) \neq y\right).$$

Duo to the unobserved counterfactual events in the observational study, we need to identify those counterfactuals as statistically estimated quantities. To this end, we extend the results in Theorem 9.2.14 of Pearl (2000) [28] for rationalization as follows, proved in Appendix A.3.

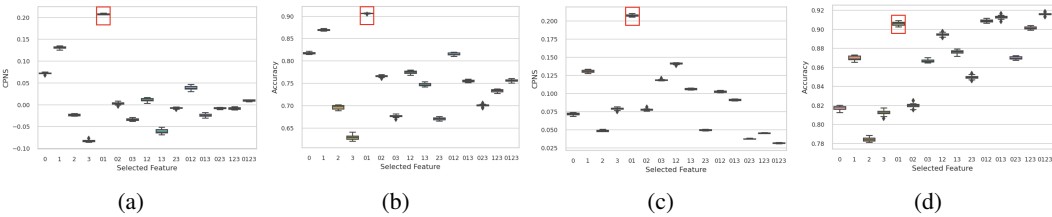

| (a) | (b) | (c) | (d) |

Figure 3: (a) and (b) show CPNS and accuracy on out-of-distribution test datasets; (c) and (d) show CPNS and accuracy on in-distribution test datasets. The red square covers the values of true rationales.

**Theorem 2.1.** *Suppose the causal graph in Figure 2 and assumptions in* (4)*,* (5)*, and* (6) *hold, then*

$$\text{CPNS}_{\{Z_j=z_j, \boldsymbol{X}=\boldsymbol{x}\}, Y=y|\boldsymbol{Z}_{-j}=\boldsymbol{z}_{-j}} = P(Y=y \mid Z_j=z_j, \boldsymbol{Z}_{-j}=\boldsymbol{z}_{-j}, \boldsymbol{X}=\boldsymbol{x}) - P(Y=y \mid Z_j \neq z_j, \boldsymbol{Z}_{-j}=\boldsymbol{z}_{-j}, \boldsymbol{X}=\boldsymbol{x}).$$

The above definition and calculation mainly focus on a single data point. In practice, we are often interested in selecting necessary and sufficient rationales for the whole dataset. Hence, we define an overall CPNS with a $\log$ transformation over the selected rationales as follows.

**Definition 2.3.** *The* $\log$ *overall* CPNS *over rationales of the whole dataset is defined as:*

$$\log \text{OCPNS} = \sum_{i=1}^{n} \sum_{j \in r_i} \frac{1}{|r_i|} \log \text{CPNS}_{\{Z_j=z_j, \boldsymbol{X}=\boldsymbol{x}\}, Y=y|\boldsymbol{Z}_{-j}=\boldsymbol{z}_{-j}},$$

*where $n$ is sample size, $r_i$ is the index set for the selected rationales of the $i$th data point, and $|r_i|$ is the number of selected rationales.*

The above definition is utilized as one component of our objective function in our main algorithm.

**Toy Example of Using POC.** We further use a toy example to demonstrate why CPNS is useful for out-of-distribution (OOD) generalization. Suppose there is a dataset of sequences, where each sequence can be represented as $\boldsymbol{X} = (X_1, \dots, X_l)$ with a equal length as $l = 4$ and a binary label $Y$. Here, we set $\{X_1, X_2\}$ are true rationale features and $\{X_3, X_4\}$ are irrelevant/spurious features. Details of simulation design are in Appendix A.4. The results of CPNS and accuracy measured in OOD and in-distribution (IND) test datasets are shown in Figure 3 over 10 replications. It can be seen that true rationales ($\{X_1, X_2\}$) yield the highest scores of CPNS and accuracy in OOD setting, and the highest $\log$ CPNS with high accuracy in IND setting. This motivates us to identify true rationales by maximizing the score of CPNS, as detailed in the next section.

## 3 Learning Necessary and Sufficient Rationale

In this section, we propose to learn necessary and sufficient rationales through the proposed $\log$ OCPNS. Our model framework consists of a selector $g_\theta(\cdot)$ and a predictor $h_\phi(\cdot)$ as standard

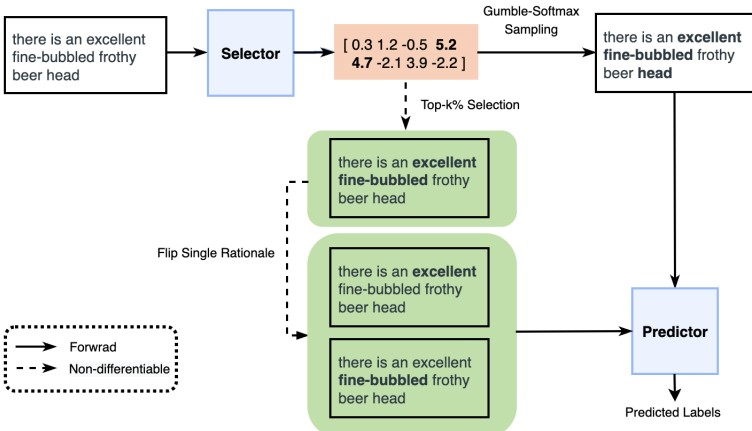

Figure 4: Model framework on learning necessary and sufficient rationale. Traditional rationalization approach only contains forward operations. In our causal rationalization, we add the causal component (highlighted by green) to incorporate non-differentiable operations and lead parallel training.

in the traditional rationalization approach, where $\theta$ and $\phi$ denote their parameters. We can get the selection $\boldsymbol{Z} = g_\theta(\boldsymbol{X})$ and fed it into predictor to get $Y = h_\phi(\boldsymbol{Z} \odot \boldsymbol{X})$ as shown in Figure 2 and Figure 4. One main difference between causal rationale and original rationale is that, we generate a series of counterfactual selections by flipping each dimension of the selection $\boldsymbol{Z}$ we obtained from the selector. Then we fed raw rationale with new counterfactual rationale into our predictor to make predictions. Considering the considerable cost of obtaining reliable rationale annotations from human, we only focus on unsupervised settings. Our goal is to make the selector create selection and rationale with the property of necessity and sufficiency and our predictor can simultaneously provide accurate predictions given such rationale. Hence we expect $P(Y = y_i \mid Z_j = z_{i,j}, \boldsymbol{Z}_{-j} = \boldsymbol{z}_{i,-j}, \boldsymbol{X} = \boldsymbol{x}_i)$ to be large while $\mathrm{P}(Y = y_i \mid Z_{i,j} \neq z_{i,j}, \boldsymbol{Z}_{i,-j} = \boldsymbol{z}_{i,-j}, \boldsymbol{X} = \boldsymbol{x}_i)$ to be small. Hence their difference, i.e., the empirical estimation of OCPNS, will be large to reflect the necessary and sufficient components. Notice that the conditional probabilities here are all estimated by the same predictor. The reason is that our causal constraints encourage necessary and sufficient rationales, which can be reflected through the prediction. For simplicity, we only consider the case where $Y$ is a binary label.

**Challenge of Directly Optimizing OCPNS.** The proposed OCPNS can be seen as a weighted sum of the difference of two estimated probabilities; there are two potential challenges if we want to directly optimize it. Firstly, many current deep learning classification networks are poorly calibrated meaning that the estimated probability of the predicted label is overestimated hence those scores are not reliable probability estimation [13, 25, 19]. Optimizing OCPNS during training adds more randomness and doesn't achieve our goal. Secondly, the difference of two probabilities sometimes can be negative when our monotonicity assumption is violated during initial training steps. In this case, we can't take the logarithm of OCPNS.

To address these computational issues, we consider the following objective function to simultaneously estimate POC from observed data and train the causal rational model that yields high prediction accuracy,

$$\min_{\theta,\phi} \mathcal{L} = \min_{\theta,\phi} \sum_{i=1}^{n} L(y_i, h_\phi(\boldsymbol{z}_i \odot \boldsymbol{x}_i)) + \mu \underbrace{\left\{ \sum_{i=1}^{n} L(y_i, h_\phi(\boldsymbol{z}_i' \odot \boldsymbol{x}_i)) - \sum_{i=1}^{n} \sum_{j=1}^{r_i} \frac{L(y_i, h_\phi(\boldsymbol{z}_{i(j)}' \odot \boldsymbol{x}_i))}{r_i} \right\}}_{\text{Causality Constraint}} + \lambda \sum_{i=1}^{n} \delta(\boldsymbol{z}_i), \quad (3)$$

where $L(\cdot, \cdot)$ defined as the cross-entropy loss, $\delta(\cdot)$ is the sparsity penalty to control sparseness of rationales, $\lambda$ and $\mu$ are the tuning parameters, $\boldsymbol{z}_i$ and $\boldsymbol{z}_i'$ are obtained through $g_\theta(\boldsymbol{x}_i)$ but under different modes. More specifically, $\boldsymbol{z}_i$ is obtained under usual training mode (Gumble-Softmax sampling) and $\boldsymbol{z}_i'$ is obtained under test mode (top $k\%$ selection). And $\boldsymbol{z}_{i(j)}'$ is obtained by flipping the $j$th dimension of $\boldsymbol{z}_i'$. Considering CPNS is the difference of two probability, to maximize it, we would like to increase the first probability and decrease the second one. Hence we can transform the difference of two probabilities of two log negative loss likelihood. It can be seen that this objective function aims to achieve the same goal as OCPNS.

The algorithm we developed to solve (3) is summarized in Algorithm 1.

# 4 Experiments and Results

## 4.1 Datasets

**Beer Review Data.** The Beer review dataset [26] is a multi-aspect sentiment analysis dataset with token-level rationales annotated by domain experts. We use its appearance, aroma, and palate aspects. **Geographic Atrophy (GA) Data.** The GA data is the advanced form of age-related macular degeneration which affects more than 5 million people worldwide and can lead to irreversible loss of vision if left untreated. Early identification of patients who are potentially developing into GA is thus critical. More details about datasets and pre-processing procedures can be found in Appendix A.5.1.

## 4.2 Synthetic Experiments

We compare our method with VIB [27] (benchmark) and the full-text model (i.e., using the full text to predict the label) by two synthetic experiments using the Beer review data. We utilize the same sparse constraint in VIB, and thus the comparison is also an ablation study to verify the usefulness of our causal module. Implementation details are provided in Appendix A.6.

**Beer-Spurious**. We include strong spurious correlation into the dataset by randomly appending spurious punctuation at the beginning of the sentence. See details in Appendix A.5.2. Table 1

**Algorithm 1** Causal Rationale

---

**Require:** Training dataset $\mathcal{D} = \{(\boldsymbol{x}_i, y_i)\}_{i=1}^N$, where $\boldsymbol{x}_i \in \mathcal{R}^d$, learning rate $\alpha$
  Initialize the parameters of selector $g_\theta(\cdot)$, predictor $h_\phi(\cdot)$, where $\theta$ and $\phi$ denote the parameters
  **while** not converge **do**
    Sample a batch $\{(\boldsymbol{x}_i, y_i)\}_{i=1}^n$ from $\mathcal{D}$
    Generate selections $\mathcal{S} = \{\boldsymbol{z}_i\}_{i=1}^n$ through Gumbel-Softmax sampling
    Generate selections $\hat{\mathcal{S}} = \{\boldsymbol{z}_i'\}_{i=1}^n$ through top $k\%$ selection
    **for** $i = 1, \ldots, n$ **do**
      **for** $j = 1, \ldots, r_i$ **do**
        Generate counterfactual selections $\boldsymbol{z}_{i(j)}'$ by flipping the rationale at the $j$th dimension of
        the initial selection $\boldsymbol{z}_i'$
      **end for**
    **end for**
    Get a new batch of selections $\tilde{\mathcal{S}} = \{\boldsymbol{z}_{i(j)}'\}_{i=1,j=r_i}^{i=n,j=d}$ and set $\mathcal{S}_{\text{all}} = \mathcal{S} \bigcup \hat{\mathcal{S}} \bigcup \tilde{\mathcal{S}}$
    Compute $\mathcal{L}$ via Eq(3) by using $\mathcal{S}_{\text{all}}$ and $\mathcal{D}$
    Update parameters $\theta = \theta - \alpha \nabla_\theta \mathcal{L}; \phi = \phi - \alpha \nabla_\phi \mathcal{L}$
  **end while**

---

Table 1: Results of synthetic experiments on the Beer-Spurious dataset. 'Acc' and 'F1' indicate the prediction accuracy and token-level F1 score of rationale selection, respectively.

| Methods | Aroma | | Palate | |
| --- | --- | --- | --- | --- |
| | Acc | F1 | Acc | F1 |
| Causal | 85.10($\pm$0.66) | 38.79($\pm$1.23) | 78.70($\pm$1.21) | 27.88($\pm$1.74) |
| VIB | 83.20($\pm$2.01) | 28.68($\pm$1.75) | 75.90($\pm$0.73) | 21.55($\pm$1.92) |
| Full Text | 89.40($\pm$1.66) | - | 82.30($\pm$0.87) | - |

Table 2: Results of synthetic experiments on the Beer-Causal dataset.

| Methods | Aroma | | Palate | |
| --- | --- | --- | --- | --- |
| | Acc | F1 | Acc | F1 |
| Causal | 86.10($\pm$0.37) | 33.52($\pm$2.19) | 80.70($\pm$0.68) | 29.18($\pm$1.39) |
| VIB | 83.10($\pm$1.62) | 26.70($\pm$1.66) | 77.60($\pm$1.46) | 20.78($\pm$1.53) |
| Full Text | 89.60($\pm$1.07) | - | 82.70($\pm$1.21) | - |

summarizes the results. It can be seen that our causal rationalization achieves a comparably good performance compared to the full-text model, and more importantly, performs consistently better than VIB w.r.t. prediction accuracy and selecting rationales by human annotations. Thus, our method is more robust when handling spurious correlation and shows better generalization performance.

**Beer-Causal**. In the second setting, we test whether our method can find true causal rationales, by randomly appending causal punctuation at the beginning of the sentence. See details in Appendix A.5.2. As summarized in Table 2, our causal rationalization obtains both higher accuracy and F1 score than VIB, similar to results in the Beer-Spurious setting. This indicates our approach can better capture human annotated rationales and appended causal rationales.

### 4.3 Real Data Experiments

We evaluate our method on two real datasets, Beer review and GA data. Our method achieves a consistently better performance than VIB on all aspects of Beer review and GA data. For Beer review dataset, especially in causal and noise scenarios, our method can not only identify causal tokens and is also robust to avoid spurious selections which matches our expectation that causal rationalization would be more robust.

In addition, for GA, it can be seen that rationalization approaches could provide better prediction performance than using either the baseline model or the full-text model. We further examine the generated rationales on GA patients with rationalization methods. We observe that our causal rationalization could provide better clinical meaningful explanations, with detailed examples provided in Appendix A.5.3.

Table 3: Results on the Beer review dataset. Causal rationalization performs the best in all aspects.

| Methods | Appearance | | Aroma | | Palate | |
|---------|------------|------|-------|------|--------|------|
| | Acc | F1 | Acc | F1 | Acc | F1 |
| Causal | 92.70($\pm$1.50) | 27.40($\pm$1.79) | 87.20($\pm$0.98) | 38.97($\pm$4.14) | 77.30($\pm$2.09) | 22.56($\pm$2.60) |
| VIB | 92.20($\pm$1.08) | 21.53($\pm$1.00) | 83.50($\pm$1.48) | 30.10($\pm$0.95) | 73.70($\pm$2.40) | 19.80($\pm$3.57) |
| Full Text | 95.70($\pm$0.60) | - | 89.90($\pm$0.37) | - | 83.10($\pm$0.20) | - |

Table 4: Results on the GA data. Both rationalization approaches beat the full-text model.

| | Causal | VIB | Full Text |
|---|--------|-----|-----------|
| Area Under the ROC Curve | 86.20($\pm$0.25) | 86.18($\pm$0.31) | 84.79($\pm$0.18) |

In summary, our method performs uniformly better than the benchmark on the Beer review data in all cases. For the GA data, our method shows a comparably better performance as VIB and provides more clinically meaningful interpretations. These justify the value of our causal rationalization.

## 5 Conclusion

In this work, we propose a novel rationalization approach to find causal interpretations for sentiment analysis tasks. We formally define non-spuriousness and efficiency of rationales from causal inference perspective and propose a practically useful algorithm. Moreover we show superior performance of our causality-based rationalization method real datasets with extensive experiments compared to state-of-the-art methods. One limitation of our method is that we make the monotonicity assumption, which is strong and can always be violated in real data. For future work, we would try to estimate the lower bound of CPNS to remove this assumption. Another promising direction is extending causal rationales to free-text rationale, which may require adjusting the events of probability of causation to align with the generation task.

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

# A   Appendix

## A.1   Related Work

**Selective Rationalization.** Selective rationalization was firstly introduced in [23] and now becomes an important model for interpretability, especially in the NLP domain [3, 4, 39, 8, 14, 9, 27, 21, 34, 34, 10, 7]. Yu et al. (2019) [39] proposed an introspective model consists of 3-players to address challenges under the co-operative setting with only the selector and the predictor. Chang et al. (2019) [8] extended the model to extract class-wise explanations. During the training phase, the initial design of the framework was not end-to-end because the sampling from these selectors were not differentiable. To address this issue, some later works adopted differentiable sampling, like gumble-softmax, other rumble-softmax, or other reparameterization tricks [see e.g., 4, 33, 18]. Paranjape al. (2020) [27] derived a sparsity-inducing objective by using Information Bottleneck and thus can explicit control sparsity of the selected rationales. One recent work [9] aimed to find the invariant rationales across multiple environments. One thing to notice is that causal definitions, including sufficiency and necessity of features were mentioned in some previous work [32, 14, 7]. However, those definitions were mainly proposed from an empirical perspective which might lack theoretical guarantees. Instead, we focus on more rigorous definitions of sufficiency and necessity from the angle of causal inference.

**Sufficiency and Necessity with Interpretable Machine Learning (XAI).** Sufficiency and necessity can be regarded as the fundamentals of XAI since they are the building blocks of all successful explanations [37]. Recently, many researchers have started to incorporate these properties into their models. Ribeiro et al. [30] proposed Anchors to find features that are sufficient to preserve current classification results. Dhurandhar et al. [15] developed an autoencoder framework to find pertinent positive (sufficient) and pertinent negative (non-necessary) features which can preserve the current results. Zhang et al. [41] considered an approach to explain a neural network by generating minimal, stable, and symbolic corrections to change model outputs. Yet, sufficiency and necessity shown in above methods are not defined from a causal perspective. There are a few works [2, 17, 37, 5] define the two properties with causal notations by using Pearl's POC to provide causal interpretations. But they are only designed for post-hoc analysis rather than developing a new model. Wang and Jordan (2021) [36] is the most relevant work to us to the best of knowledge. In their work, they formally defined sufficiency and necessity for learned representation with POC and showed identification of those counterfactual quantities. Finally, they incorporated those causal inspired constraints into the training process to obtain non-spurious and efficient representations. In our work, we generalize these concepts to rationalization and enable our selected rationales to satisfy the desiderata.

## A.2   Assumptions

**Assumption A.1.**

- *Consistency:*
$$Y(\boldsymbol{Z} = \boldsymbol{z}, \boldsymbol{X} = \boldsymbol{x}) = Y. \tag{4}$$

- *Ignorability:*
$$\{Y(Z_j = z_j, \boldsymbol{Z}_{-j} = \boldsymbol{z}_{-j}, \boldsymbol{X} = \boldsymbol{x}), Y(Z_j \neq z_j, \boldsymbol{Z}_{-j} = \boldsymbol{z}_{-j}, \boldsymbol{X} = \boldsymbol{x})\} \perp \boldsymbol{Z}. \tag{5}$$

- *Monotonicity (with $\boldsymbol{X}$, $Y$ is monotonic relative to $Z_i$):*
$$\{Y(Z_j \neq z_j, \boldsymbol{Z}_{-j} = \boldsymbol{z}_{-j}, \boldsymbol{X} = \boldsymbol{x}) = y\} \wedge \{Y(Z_j = z_j, \boldsymbol{Z}_{-j} = \boldsymbol{z}_{-j}, \boldsymbol{X} = \boldsymbol{x}) \neq y\} = \textit{False}, \tag{6}$$
*where $\wedge$ is the logical operation AND. For Two events A and B, $A \wedge B$ =True if $A = B$ =True, $A \wedge B$ =False otherwise.*

Here, the first assumption implies that the potential label of text under the given selection is the label that would be observed for that text. The second assumption in causal inference usually means no unmeasured confounders, which is automatically satisfied under randomized trials. For observational studies, we rely on domain experts to include as many features as possible to guarantee this assumption. In rationalization, it means our text already contains all information. For monotonicity assumption, it indicates that a change on the wrong selection will not, under any circumstance, make $Y$ change to the true label. In other words, that true selection will increase the likelihood of the true label.

### A.3 Proof of Theorem 2.1

**Proof:** Let's denote the logical operators *and*, *or* as $\wedge, \vee$, respectively.

Firstly we know

$$\{Y\left(Z_i = z_i, \boldsymbol{Z}_{-i} = \boldsymbol{z}_{-i}, \boldsymbol{X} = \boldsymbol{x}\right) = y\} \vee \{Y\left(Z_i = z_i, \boldsymbol{Z}_{-i} = \boldsymbol{z}_{-i}, \boldsymbol{X} = \boldsymbol{x}\right) \neq y\} = \text{True}. \quad (7)$$

Then we have

$$
\begin{aligned}
&\{Y\left(Z_j \neq z_j, \boldsymbol{Z}_{-j} = \boldsymbol{z}_{-j}, \boldsymbol{X} = \boldsymbol{x}\right) = y\} \\
\overset{(7)}{=}\ &\{Y\left(Z_j \neq z_j, \boldsymbol{Z}_{-j} = \boldsymbol{z}_{-j}, \boldsymbol{X} = \boldsymbol{x}\right) = y\} \\
&\wedge \left[\{Y\left(Z_j = z_j, \boldsymbol{Z}_{-j} = \boldsymbol{z}_{-j}, \boldsymbol{X} = \boldsymbol{x}\right) = y\} \vee \{Y\left(Z_j = z_j, \boldsymbol{Z}_{-j} = \boldsymbol{z}_{-j}, \boldsymbol{X} = \boldsymbol{x}\right) \neq y\}\right] \\
=\ &\left[\{Y\left(Z_j \neq z_j, \boldsymbol{Z}_{-j} = \boldsymbol{z}_{-j}, \boldsymbol{X} = \boldsymbol{x}\right) = y\} \wedge \{Y\left(Z_j = z_j, \boldsymbol{Z}_{-j} = \boldsymbol{z}_{-j}, \boldsymbol{X} = \boldsymbol{x}\right) = y\}\right] \\
&\vee \left[\{Y\left(Z_j \neq z_j, \boldsymbol{Z}_{-j} = \boldsymbol{z}_{-i}, \boldsymbol{X} = \boldsymbol{x}\right) = y\} \wedge \{Y\left(Z_j = z_j, \boldsymbol{Z}_{-j} = \boldsymbol{z}_{-j}, \boldsymbol{X} = \boldsymbol{x}\right) \neq y\}\right] \\
\overset{(6)}{=}\ &\left[\{Y\left(Z_j \neq z_j, \boldsymbol{Z}_{-j} = \boldsymbol{z}_{-j}, \boldsymbol{X} = \boldsymbol{x}\right) = y\} \wedge \{Y\left(Z_j = z_j, \boldsymbol{Z}_{-j} = \boldsymbol{z}_{-j}, \boldsymbol{X} = \boldsymbol{x}\right) = y\}\right], (8)
\end{aligned}
$$

where we use the monotonicity assumption in (6).

Also, we know

$$\{Y\left(Z_i \neq z_i, \boldsymbol{Z}_{-i} = \boldsymbol{z}_{-i}, \boldsymbol{X} = \boldsymbol{x}\right) = y\} \vee \{Y\left(Z_i \neq z_i, \boldsymbol{Z}_{-i} = \boldsymbol{z}_{-i}, \boldsymbol{X} = \boldsymbol{x}\right) \neq y\} = \text{True}. \quad (9)$$

Then we can get

$$
\begin{aligned}
&\{Y\left(Z_j = z_j, \boldsymbol{Z}_{-j} = \boldsymbol{z}_{-j}, \boldsymbol{X} = \boldsymbol{x}\right) = y\} \\
\overset{(9)}{=}\ &\{Y\left(Z_j = z_j, \boldsymbol{Z}_{-j} = \boldsymbol{z}_{-j}, \boldsymbol{X} = \boldsymbol{x}\right) = y\} \\
&\wedge \left[\{Y\left(Z_j \neq z_j, \boldsymbol{Z}_{-j} = \boldsymbol{z}_{-j}, \boldsymbol{X} = \boldsymbol{x}\right) = y\} \vee \{Y\left(Z_j \neq z_j, \boldsymbol{Z}_{-j} = \boldsymbol{z}_{-j}, \boldsymbol{X} = \boldsymbol{x}\right) \neq y\}\right] \\
=\ &\left[\{Y\left(Z_j = z_j, \boldsymbol{Z}_{-j} = \boldsymbol{z}_{-j}, \boldsymbol{X} = \boldsymbol{x}\right) = y\} \wedge \{Y\left(Z_j \neq z_j, \boldsymbol{Z}_{-j} = \boldsymbol{z}_{-j}, \boldsymbol{X} = \boldsymbol{x}\right) = y\}\right] \\
&\vee \left[\{Y\left(Z_j = z_j, \boldsymbol{Z}_{-j} = \boldsymbol{z}_{-i}, \boldsymbol{X} = \boldsymbol{x}\right) = y\} \wedge \{Y\left(Z_j \neq z_j, \boldsymbol{Z}_{-j} = \boldsymbol{z}_{-j}, \boldsymbol{X} = \boldsymbol{x}\right) \neq y\}\right] \\
\overset{(8)}{=}\ &\{Y\left(Z_j \neq z_j, \boldsymbol{Z}_{-j} = \boldsymbol{z}_{-j}, \boldsymbol{X} = \boldsymbol{x}\right) = y\} \\
&\vee \left[\{Y\left(Z_j = z_j, \boldsymbol{Z}_{-j} = \boldsymbol{z}_{-i}, \boldsymbol{X} = \boldsymbol{x}\right) = y\} \wedge \{Y\left(Z_j \neq z_j, \boldsymbol{Z}_{-j} = \boldsymbol{z}_{-j}, \boldsymbol{X} = \boldsymbol{x}\right) \neq y\}\right].
\end{aligned}
$$

$$(10)$$

Based on the consistency assumption in (4), we either have $\{Y\left(Z_j \neq z_j, \boldsymbol{Z}_{-j} = \boldsymbol{z}_{-j}, \boldsymbol{X} = \boldsymbol{x}\right) = y\}$ or $\{Y\left(Z_j \neq z_j, \boldsymbol{Z}_{-j} = \boldsymbol{z}_{-j}, \boldsymbol{X} = \boldsymbol{x}\right) \neq y\}$ holds. Therefore, we know the two events in the last line of (10) are disjoint and further take the probability on both sides to get:

$$
\begin{aligned}
&P\left(Y\left(Z_j = z_j, \boldsymbol{Z}_{-j} = \boldsymbol{z}_{-j}, \boldsymbol{X} = \boldsymbol{x}\right) = y\right) \\
=\ &P\left(Y\left(Z_j \neq z_j, \boldsymbol{Z}_{-j} = \boldsymbol{z}_{-j}, \boldsymbol{X} = \boldsymbol{x}\right) = y\right) \\
&+ P\left(Y\left(Z_j = z_j, \boldsymbol{Z}_{-j} = \boldsymbol{z}_{-i}, \boldsymbol{X} = \boldsymbol{x}\right) = y, Y\left(Z_j \neq z_j, \boldsymbol{Z}_{-j} = \boldsymbol{z}_{-j}, \boldsymbol{X} = \boldsymbol{x}\right) \neq y\right),
\end{aligned}
$$

$$(11)$$

where the last term is exactly $\text{CPNS}_{\{Z_j = z_j, \boldsymbol{X} = \boldsymbol{x}\}, Y = y | \boldsymbol{Z}_{-j} = \boldsymbol{z}_{-j}}$ which we want to identify.

Finally with our ignorability assumption (5) we get:

$$
\begin{aligned}
&\text{CPNS}_{\{Z_j = z_j, \boldsymbol{X} = \boldsymbol{x}\}, Y = y | \boldsymbol{Z}_{-j} = \boldsymbol{z}_{-j}} \\
=\ &P\left(Y\left(Z_j = z_j, \boldsymbol{Z}_{-j} = \boldsymbol{z}_{-j}, \boldsymbol{X} = \boldsymbol{x}\right) = y\right) - P\left(Y\left(Z_j \neq z_j, \boldsymbol{Z}_{-j} = \boldsymbol{z}_{-j}, \boldsymbol{X} = \boldsymbol{x}\right) = y\right) \\
\overset{(5)}{=}\ &P(Y = y \mid Z_j = z_j, \boldsymbol{Z}_{-j} = \boldsymbol{z}_{-j}, \boldsymbol{X} = \boldsymbol{x}) - P(Y = y \mid Z_j \neq z_j, \boldsymbol{Z}_{-j} = \boldsymbol{z}_{-j}, \boldsymbol{X} = \boldsymbol{x}).
\end{aligned}
$$

### A.4 Toy Example

The process of generating such dataset is described below. Firstly, we generate rationale features $\{X_1, X_2\}$ following a bivariate normal distribution with positive correlations. To create spurious correlation, we generate irrelevant features $\{X_3, X_4\}$ by using mapping functions which map $\{X_1, X_2\}$ to $\{X_3, X_4\}$, thus irrelevant features are highly correlated with rationale features. There are datasets:

the training dataset, the in-distribution test data and the out-of-distribution test data. The training data $\left\{\boldsymbol{x}_i^{\text{train}}\right\}_{i=1}^{n_{\text{train}}}$ and the in-distribution test data $\left\{\boldsymbol{x}_i^{\text{test-in}}\right\}_{i=1}^{n_{\text{test-in}}}$ follows the same generation process, but for out-of-distribution test data $\left\{\boldsymbol{x}_i^{\text{test-out}}\right\}_{i=1}^{n_{\text{test-out}}}$, we modify the mapping function to create a different distribution of the features. Then we make the label $Y$ only depends on $(X_1, X_2)$. This is equivalent to assume that all the rationales are on the same position and the purpose is to simplify the explanations. For a single $\boldsymbol{x}_i$, we simulate $\mathrm{P}(y_i = 1|\boldsymbol{x}_i) = \pi(\boldsymbol{x}_i)$ below:

$$\pi(\boldsymbol{x}_i) = \frac{1}{1 + e^{-(\beta_0 + \beta_1 x_{i1} + \beta_2 x_{i2})}}.$$

Then we use threshold value 0.5 to categorize the data into one of two classes: $y_i = 1$ if $\pi \geq 0.5$ and $y_i = 0$ if $\pi \leq 0.5$. Since the dataset includes 4 features, there are 15 combinations of the rationale (we ignore the rationale containing no features. For each rationale, we would fit a logistic regression model by using only selected features and refit a new logistic regression model with subset features to calculate CPNS. In practice monotonicity can be violated (removing one of the selected rational increases the predicted probability) and it can cause problems when we use log transform. Hence we calculate a surrogate of CPNS:

$$\mathrm{CPNS}_{\{Z_j = z_j, \boldsymbol{X} = \boldsymbol{x}\}, Y = y | \boldsymbol{Z}_{-j} = \boldsymbol{z}_{-j}} = P\left(Y\left(Z_j = z_j, \boldsymbol{Z}_{-j} = \boldsymbol{z}_{-j}, \boldsymbol{X} = \boldsymbol{x}\right) = y, Y\left(Z_j \neq z_j, \boldsymbol{Z}_{-j} = \boldsymbol{z}_{-j}, \boldsymbol{X} = \boldsymbol{x}\right) \neq y\right) + 1.$$

In our simulation, we set $n_{\text{train}} = n_{\text{test-in}} = n_{\text{test-out}} = 20000$, $\beta_0 = 1$, $\beta_1 = 0.5$ and $\beta_2 = 1$.

## A.5 Experiments

### A.5.1 Datasets

**Beer Review Data.** Following the same evaluation protocol of some previous works [see e.g., 3, 39, 9, 10], we convert the their original scores which are in the scale of $[0, 1]$ into binary labels. Specifically, reviews with ratings $\leq 0.4$ are labeled as negative and those with $\geq 0.6$ are labeled as positive. We follow the same train/dev/test split as [10] and it is summarized in Table 5. To make computation more feasible, except the raw dataset, we create a short-text version of the dataset by filtering the texts over length 120. Table 6 summarizes the statistics of the Beer review dataset. This dataset contains Token-level annotations from humans. Hence we use accuracy and token F1 to evaluate task performance and human annotated rationales agreement.

Table 5: The split of the dataset.

| Short | Train | Dev | Test |
|---|---|---|---|
| Beer (Look) | 15932 | 3757 | 200 |
| Beer (Aroma) | 14085 | 2928 | 200 |
| Beer (Palate) | 9592 | 2294 | 200 |

Table 6: Dataset details, with rationale length ratios included for datasets where they are available.

| Short | Len | Rationale(%) | N |
|---|---|---|---|
| Beer (Look) | 88.94 | 19.2 | 19889 |
| Beer (Aroma) | 89.92 | 15.9 | 17033 |
| Beer (Palate) | 90.72 | 12.7 | 12086 |

**GA Data.** The proprietary GA dataset used in this study include the medical claim records (diagnosis, prescriptions, and procedures) of 329,023 patients who are diagnosed as GA during 2018 to 2021 in US, as well as the records of additional 991,946 patients who have at least one of GA risk factors. We are tasked to utilize the medical claim data to find high-risk GA patients and reveal important clinical indications using the causal rationale framework. For this dataset, we use the area under the ROC curve (AUC) to evaluate task performance. This dataset doesn't provide human annotations because it requires a huge amount of time and money to hire domain experts to annotate such a big dataset.

### A.5.2 Synthetic Dataset

**Beer-Spurious**. The first setting is considered to include strong spurious correlation into the dataset. We follow a similar setup in [9, 40] to append punctuation "," and "." at the beginning of the first

Table 7: The split of the dataset.

| Short | Train | Dev | Test | Total | Len |
|-------|-------|-----|------|-------|-----|
| GA | 20000 | 10000 | 10000 | 40000 | 122.31 |

Causal                   VIB

'POLYP OF CORPUS UTERI',

'DEFICIENCY OF OTHER SPECIFIED B GROUP VITAMINS',

'UNSPECIFIED MACULAR DEGENERATION',

'EXUDATIVE AGE-RELATED MACULAR DEGENERATION',

'PRESENCE OF INTRAOCULAR LENS',

'NONEXUDATIVE AGE-RELATED MACULAR DEGENERATION',

'UNSPECIFIED MACULAR DEGENERATION',

'LESION OF SCIATIC NERVE, LEFT LOWER LIMB',

'OBESITY, UNSPECIFIED',

'EXPOSURE TO OTHER SPECIFIED FACTORS, SUBSEQUENT ENCOUNTER',

'POLYP OF CORPUS UTERI'

'UNSPECIFIED MACULAR DEGENERATION',

'EXUDATIVE AGE-RELATED MACULAR DEGENERATION',

'NONEXUDATIVE AGE-RELATED MACULAR DEGENERATION',

'UNSPECIFIED MACULAR DEGENERATION',

'OBESITY, UNSPECIFIED',

'AGE-RELATED OSTEOPOROSIS WITHOUT CURRENT PATHOLOGICAL FRACTURE',

'PAIN IN LEFT WRIST',

'EXPOSURE TO OTHER SPECIFIED FACTORS, SUBSEQUENT ENCOUNTER',

'EXPOSURE TO OTHER SPECIFIED FACTORS, SUBSEQUENT ENCOUNTER',

'POLYP OF CORPUS UTERI'

Figure 5: An example of generated rationales of positive patients. Left is from causal rationalization and right is from baseline VIB. Red and blue indicates the main difference between two sets of rationales

sentence with the following distributions:

$$P(\text{ append ","} \mid Y = 1) = P(\text{ append "."} \mid Y = 0) = \alpha_1,$$
$$P(\text{ append "."} \mid Y = 1) = P(\text{ append ","} \mid Y = 0) = 1 - \alpha_1.$$

Here we set $\alpha_1 = 0.8$. Intuitively, since the first sentence now contains the appended punctuation with strong spurious correlation, we expect the usual rationalization approach to capture such a clue and our method can identify the spurious feature and avoid selecting it.

**Beer-Causal**. Firstly we randomly append either punctuation "," or "." at the beginning of the first sentence with probability $P(\text{ append ","}) = P(\text{ append "."}) = 0.5$. Then we regard the punctuation as the causal rational which can influence the label:

$$P(Y_{\text{new}} = 0 \mid Y_{\text{old}} = 1, ",") = P(Y_{\text{new}} = 1 \mid Y_{\text{old}} = 0, ".") = \alpha_2.$$

Here we set $\alpha_2 = 0.4$. Intuitively, since the first sentence now contains the appended punctuation with mild causal correlation, we expect the usual rationalization approach to miss such a clue and our method can identify the causal feature and select it.

### A.5.3 Extracted Interpretations for GA Patients

We visualize the rationales of one GA patient from the claim data extracted using both our method and the baseline in Figure 5. Overall the two methods extracted similar rationales: they all identified "AGE-RELATED MACULAR DEGENERATION" which are the diagnoses of early stage GA, as well as "OBESITY, UNSPECIFIED" which is a known GA risk factor.

What our method excels is 1) it identified "PRESENCE OF INTRAOCULAR LENS" which indicates history of eye surgery for treating myopia or cataracts and is highly pertinent to GA progression, and 2) it avoids "PERSONS ENCOUNTERING HEALTH SERVICES IN OTHER SPECIFIED CIRCUMSTANCES", which is a generic billing code and clinically irrelevant.

### A.6 Implementation Details

For the Beer review data, we use two BERT-base-uncased as the selector and the predictor components for rationalization approaches, and one BERT-base for the full-text model. Those modules are initialized with pre-trained Bert. For the GA data, we use the same architecture and the only

difference is we replace the word embedding matrix with a randomly initialized health diagnosis code embedding and the embedding is trained jointly with other modules.

In our experiments, we utilize the Sparse IB constraints for rationalization approaches from [27]. We didn't include a continuity constraint for two reasons. Firstly, starting with less tuning parameters is more efficient to verify the usefulness of our causal module. Secondly, for the GA data, we don't expect the rationales of medical claim records to be continuous spans because continuous explanations don't help and are even unrealistic.

For all experiments, we utilize a batch size of 256 and choose the learning rate $\alpha \in \{1e-5, 5e-4, 1e-4\}$. For two branches in our framework, differentiable Gumbel–Softmax and hard top-$k\%$ are utilized to compute task loss and causal component loss respectively.

We train for 20 epochs for the Beer review data and 10 epochs for the GA data where the first 5 epochs are warm-up (trained without causal component). For training the causal component, we tune the values of the Lagrangian multiplier $\mu \in \{0.01, 0.1, 1\}$ and set $k = 5$. We set the temperature of Gumbel-softmax to be $0.1$. For our final evaluation, we choose top-$10\%$ tokens as the rationales. We conduct our experiments over 5 random seeds and calculate the mean and standard deviation of metrics.

