# OpenReview forum: "On Causal Rationalization"
_NeurIPS.cc/2022/Workshop/TSRML — TSRML2022_

### Official Review · Reviewer_1htN · 2022-10-14
**A Causal Perspective on Explaining Language Models**

**Overall Rating:** 6

**Summary:**

This paper deals with the problem of rationalization in NLP tasks. In particular, a method motivated by a causal perspective is developed and tested on the benchmark Beer review dataset and EHRs.

**Strengths:**

* The problem of finding important features among highly correlated variables is relevant beyond the NLP tasks and is of general interest to interpretable ML.

* The proposed method has explicit assumptions regarding the underlying generative process behind the data attempts to formalise the desiderate posed against a "good" rationale.

* The paper is well-written and features figures with concrete examples that make the material easier to follow.

**Weaknesses:**

* The experiments do not include many baselines. It would have been nice to see an empirical comparison with a more diverse set of baselines, e.g., LIME, gradient-based attributions measures, and other rationalization techniques, for instance, the work by Chrysostomou and Aletras (2022) seems quite closely related.

* In a similar vein, looking at the final loss function used for learning rationales (Eq. 3), the proposed method looks similar to the Granger-causal attentive mixtures of experts by Schwab et al. (2019).

* It is not apparent (to me) why the loss function in Eq. 3 corresponds to OCPNS.

* The paper provides no conclusion and does not discuss avenues for future work.

* The proposed technique seems to yield no significant improvement over VIB on the medical data.

* Assumptions 4-6 outline the limitations of the proposed CPNS. It would have been nice to see at least some discussion of the limitations of the proposed approach.

### References

Schwab, P., Miladinovic, D., & Karlen, W. (2019). Granger-causal attentive mixtures of experts: Learning important features with neural networks. In *Proceedings of the AAAI Conference on Artificial Intelligence* (Vol. 33, No. 01, pp. 4846-4853).

Chrysostomou, G., & Aletras, N. (2022). Flexible Instance-Specific Rationalization of NLP Models. *AAAI*.

**Overall Recommendation:**

This paper extends the classical rationalisation for language models to be able to deal with correlated tokens, treating the problem from the causal perspective. Admittedly, the selection of baselines in the experiments is limited, and it is unclear whether the loss function used in practice corresponds to the quantities described in the theoretical motivation. Since I am not an expert in NLP, it is difficult for me to evaluate the significance of the current work. However, the results appear reasonable, and the paper seems to be a good fit for the workshop thematically.

**Review Confidence:**

2: The reviewer is willing to defend the evaluation, but it is quite likely that the reviewer did not understand central parts of the paper

---

### Official Review · Reviewer_BgPw · 2022-10-14
**Review of Paper 37**

**Overall Rating:** 7

**Summary:**

This paper proposes a new method to solve critical problems of existing association-based approaches on rationalization. In particular, the authors leverage two causal desiderata, non-spuriousness and efficiency, into rationalization from the causal inference perspective. The experiment results on real-world review and medical datasets show that their method outperforms state-of-the-art methods.

**Strengths:**

* The idea of discovering causality in the rationalization area is novel. Providing a trustworthy explanation of prediction is also highly related to the topic of this workshop.

* Clear definition and solid theoretical analysis.


**Weaknesses:**

None

**Overall Recommendation:**

I recommend accepting this paper because of its novel idea and high-quality experiment.


**Review Confidence:**

4: The reviewer is confident but not absolutely certain that the evaluation is correct

---

### Decision · Program_Chairs · 2022-10-23

Accept